# Multi-Omics Nutritional Approaches Targeting Metabolic-Associated Fatty Liver Disease

**DOI:** 10.3390/genes13112142

**Published:** 2022-11-17

**Authors:** Omar Ramos-Lopez

**Affiliations:** Medicine and Psychology School, Autonomous University of Baja California, Tijuana 22390, Mexico; oscar.omar.ramos.lopez@uabc.edu.mx

**Keywords:** MAFLD, precision nutrition, nutrigenetics, nutrigenomics, nutriepigenetics, metagenomics, proteomics, metabolomics

## Abstract

Currently, metabolic-associated fatty liver disease (MAFLD) is a leading global cause of chronic liver disease, and is expected to become one of the most common indications of liver transplantation. MAFLD is associated with obesity, involving multiple mechanisms such as alterations in lipid metabolism, insulin resistance, hyperinflammation, mitochondrial dysfunction, cell apoptosis, oxidative stress, and extracellular matrix formation. However, the onset and progression of MAFLD is variable among individuals, being influenced by intrinsic (personal) and external environmental factors. In this context, sequence structural variants across the human genome, epigenetic phenomena (i.e., DNA methylation, histone modifications, and long non-coding RNAs) affecting gene expression, gut microbiota dysbiosis, and metabolomics/lipidomic fingerprints may account for differences in MAFLD outcomes through interactions with nutritional features. This knowledge may contribute to gaining a deeper understanding of the molecular and physiological processes underlying MAFLD pathogenesis and phenotype heterogeneity, as well as facilitating the identification of biomarkers of disease progression and therapeutic targets for the implementation of tailored nutritional strategies. This comprehensive literature review highlights the potential of nutrigenetic, nutriepigenetic, nutrimetagenomic, nutritranscriptomics, and nutrimetabolomic approaches for the prevention and management of MAFLD in humans through the lens of precision nutrition.

## 1. Introduction

Metabolic-associated fatty liver disease (MAFLD) was recently proposed as a more appropriate overarching term rather than nonalcoholic fatty liver disease (NAFLD) to refer to a chronic liver disease typically associated with obesity, with a more focused clinical approach [1]. The MAFLD spectrum ranges from simple steatosis (abnormal accumulation of triglycerides in the hepatocyte) to steatohepatitis (the inflammatory advanced stage of this disease), cirrhosis, and even liver cancer [2]. Currently, MAFLD is a leading global cause of chronic liver disease, and is expected to become one of the most common indications of liver transplantation [3].

MAFLD pathogenesis involves multiple mechanisms such as disruptions in lipid metabolism, insulin resistance, hyperinflammation, mitochondrial dysfunction, cell apoptosis, oxidative stress, extracellular matrix formation (fibrosis), and intestinal microbiota alterations [4]. However, the onset and progression of MAFLD is variable among individuals, being influenced by genetic, epigenetic, and food environments specific to each population [5].

In this context, a number of DNA sequence structural variants across the human genome have been associated with the susceptibility and severity of MAFLD trough interactions with nutritional factors [6]. In addition, dietary regulation of gene expression affecting MAFLD pathogenesis may involve epigenetic mechanisms including DNA methylation, histone modifications, and long non-coding RNA features [7]. Of note, gut microbiota dysbiosis is a common feature in MAFLD patients, mainly combined with nutritionally unbalanced poor diets [8]. Furthermore, metabolomic and lipidomic fingerprints related to food consumption have been identified as potential biomarkers for diagnosis, prognosis, and monitoring of MAFLD development and phenotype categorization [9].

This knowledge may contribute to gaining a deeper understanding of the molecular and physiopatological mechanisms underlying MAFLD, as well as facilitating the identification of therapeutic targets for the design of tailored nutritional strategies. This literature review provides examples of emerging nutrigenetic, nutriepigenetic, nutrimetagenomic, nutritranscriptomics, and nutrimetabolomic approaches for the prevention and management of MAFLD in humans through the lens of precision nutrition (Figure 1).

### 1.1. Nutrigenetics

The variability in MAFLD phenotypes can be influenced by the genetic background under certain nutritional conditions [10]. For example, after carbohydrate overfeeding (consuming an extra 1000 kcal/d with 98% of energy from carbohydrates over 3 weeks), an increase in liver fat content and serum triglycerides concentrations was observed in *PNPLA3**GG homozygous subjects for the rs738409 polymorphism, but not in *PNPLA3**CC carriers [11]. In agreement with these findings, carbohydrate/sugar consumption was positively associated with hepatic fat deposition only in overweight Hispanic children carrying the *PNPLA3* GG genotype [12]. Accordingly, this genotype was also related to higher degrees of hepatic steatosis in adolescents who reported drinking sweetened drinks at least once weekly [13].

Of note, liver fat accumulation and the serum levels of alanine aminotransferase were influenced by the interaction between the ratio of omega-6/omega-3 polyunsaturated fatty acid intake and the *PNPLA3* GG genotype in children and adolescents [14]. Similarly, the intakes of several dietary types of unsaturated fat (including monounsaturated fatty acids, polyunsaturated fatty acids, and omega-6 fatty acids) were differentially associated with liver fibrosis by *PNPLA3* G risk alleles [15]. Remarkably, it was demonstrated that the *PNPLA3* rs738409 variant significantly modulated the relationship between MAFLD-related fibrosis severity and the intakes of carbohydrates, omega-3 fatty acids, total isoflavones, methionine, and choline intakes in non-Hispanic whites [16]. Interestingly, a meta-analysis demonstrated that people with the GG genotype of the *PNPLA3* rs738409 variant were 105% more likely to develop MAFLD, followed by the CG heterozygotes (19% higher risk of MAFLD); conversely, those carrying the CC genotype had a 52% lower chance of presenting this outcome [17].

In addition to *PNPLA3* variations, it was evidenced that obese adolescents with the TT genotype of the *GCKR* rs1260326 polymorphism underwent higher rates of liver de novo lipogenesis after an acute carbohydrate (75 g glucose and 25 g fructose) challenge [18]. In addition, participants with the minor haplotype in the 22q13 loci (comprising *LARGE* rs240072, *RBFOX2* rs11089778, *TRIOBP* rs12628603, *PNPLA3* rs738409, and *PARVB* rs2073080 genetic variants) had a higher MAFLD risk that was exacerbated by high carbohydrate intake (75% of energy from carbohydrates) and noodle/meat-rich dietary patterns (70th percentiles) in Koreans [19]. Furthermore, high fruit intake (>2 portions/d) increased the risk of MAFLD in subjects carrying the risk alleles of at least one of the following metabolic gene polymorphisms involved in oxidative stress: *GSTT1*null*, *GSTM1*null*, *SULT1A1*2*, *CYP2E1*6*, and *CYP1A1*2A* [20]. Moreover, an increase in fish intake by one portion/week exerted an additive effect on the risk of developing MAFLD in subjects carrying the risk allele of the *TM6SF2* rs58542926 variant [21].

Some studies have also evaluated the influence of genetic polymorphisms in response to nutritional interventions in MAFLD. In this regard, the *PNPLA3* GG genotype was associated with greater liver fat reductions compared with CC homozygotes after consuming a hypocaloric (1000 kcal deficit/d) low-carbohydrate diet (20 g/d of carbohydrates) for 6 days [22]. Likewise, *PNPLA3* GG genotype carriers displayed a significant change in hepatic fat and plasma triglyceride levels at the end of an intervention with a low-omega 6:omega 3 ratio (4:1) normocaloric diet in obese youth [23]. Furthermore, it was reported that subjects carrying the T allele of the *SH2B1* rs7359397 polymorphism showed a greater decrease in liver fat content when prescribed an energy-restricted treatment (−30% of the individual’s requirements) for 6 months [24].

Concerning the use of nutritional supplements for treating MAFLD, it was demonstrated that the *PNPLA3* G allele attenuated the beneficial effect of 6 months of supplementation (silymarin 60% plus 30 IU vitamin E/d) on reducing plasma transaminases levels in MAFLD patients [25]. Findings from the WELCOME trial also revealed that the *PNPLA3* GG genotype worsened liver fat status and docosahexaenoic acid (DHA) tissue enrichment after 4 g DHA + eicosapentaenoic acid (EPA) supplementation for 15–18 months in subjects with MAFLD [26]. Comparably, the *PNPLA3* G allele led to a decreased response to DHA supplementation (250–500 mg/d) in children with MAFLD [27].

### 1.2. Nutriepigenetics

Dysregulation of epigenetic phenomena (including DNA methylation, histone modifications, and long non-coding RNA features) can alter cell phenotype and related physiological functions, leading to the onset and progression of diverse chronic diseases, including MAFLD [28].

*DNA methylation*: Of note, a nutriepigenetic analysis demonstrated that a low carbohydrate diet (37–40% of energy from carbohydrates) alone or combined with aerobic exercise over 6 months can protect against the progression of MAFLD by inducing methylation changes at the *GAB2* gene in blood [29]. In addition, a differential DNA methylation pattern at the *A2MP1* gene was identified in MAFLD participants adhering to either an isocaloric low-fat (30% of calories from fat) diet or Mediterranean low-carbohydrate (40–70 g/d of carbohydrates) diet plus 28 g walnuts/d over 18 months [30]. Moreover, in vitro analyses using HepG2 human hepatoma cells revealed that resveratrol administration attenuated glucose-induced MAFLD development through DNA methylation modification of the *NRF2* gene promoter [31].

*Histone modifications*: In vitro assays have shown that tannic acid (a hydrolysable tannin polyphenol found in many dietary plant products such as coffee, tea, cocoa, and sorghum grain) ameliorates lipid accumulation via downregulation of lipogenesis-related gene (*SREBP-1c*, *ACLY*, *FASN*, *PPARγ*) expression and inhibition of histone acetyltransferase activity in HepG2 human cells [32]. Using this same cell line, black mulberry fruit extract markedly reduced the expression of proteins associated with lipogenesis, which was attributed to suppression of total acetylated lysine as well as specific histone acetylation of proteins H3K14 and H3K27 [33]. Likewise, *Schisandra chinensis* berry extract protected against steatosis development by inhibiting histone H3K9 acetylation in oleic acid-treated HepG2 cells [34].

*Long non-coding RNA*: MicroRNAs (miRNAs) play a role in nutrient metabolism (i.e., dietary fatty acids) by modulating gene expression, with potential applications as biomarkers in metabolic diseases including MAFLD [35]. Accordingly, miR-26a potentially contributed to the regulation of fatty acid and sterol homeostasis in an in vitro cell model of MAFLD induced by palmitic acid (PA) and oleic acid (OA), which occur naturally in various animal and vegetable fats and oils [36]. Likewise, PA supply led to decreased miR-139-5p, miR-30b-5p, miR-422a, and miR-146a in human hepatocytes, in parallel with increased lipogenesis and fatty acid transport, but also led to decreased glucose metabolism and diminished fatty acid oxidation [37]. Furthermore, miR-122 [38], miR-34a [39], miR-141 [40], and miR-23b [41] influenced nutrient-induced MAFLD pathogenesis by targeting sirtuin 1 in vitro. Moreover, the long non-coding RNA-H19 promoted hepatic lipogenesis in human liver cells treated with PA/OA by directly altering the miR-130a/PPARγ axis [42]. Several vitamin D-modulated miRNAs (such as miR-27, miR-125, miR-155, miR-192, miR-223, miR-375, and miR-378) have been identified as potentially relevant to MAFLD pathogenesis, emphasizing the importance of measuring serum and hepatic miRNAs in response to vitamin D supplementation in upcoming trials [43].

### 1.3. Nutrimetagenomics

Advances in sequencing methods have allowed the identification and taxonomic characterization of microbial communities and their effects on health using genomic techniques, such as 16S rRNA amplicon or shotgun metagenomic sequencing [44]. Growing evidence supports the modulatory role of diet in human gut microbiome composition, affecting several metabolic pathways in the host [45]. In particular, it has been reported that dietary cholesterol, fiber, fat, or carbohydrates could modify the microbiome to contribute to the development of MAFLD and its accompanying liver complications [46].

This knowledge emphasizes the search for nutritional strategies improving metabolic and liver-related markers in MAFLD by restoring the homeostasis of the intestinal microbiota [47]. Interestingly, an increase in nutritional fiber (from 19 g/d to 29 g/d) resulted in reduced liver enzymes and improved hepatic steatosis in patients with MAFLD undergoing weight reduction, possibly by altering intestinal permeability [48]. Correspondingly, it has been suggested that the incorporation of prebiotics (non-digestible carbohydrates) into the diet may reduce hepatic lipids and ameliorate risk factors associated with MAFLD [49]. Accordingly, it was reported that supplementation with inulin-type fructans (ITF) prebiotics (16 g/d of inulin/oligofructose for 3 months) in obese women led to higher abundances in *Bifidobacterium* and *Faecalibacterium prausnitzii* as well as a decrease in *Bacteroides intestinalis*, *B. vulgatus*, and *Propionibacterium*, with potential implications in the management of obesity-related liver disease [50]. Furthermore, children who were overweight or obese and received oligofructose-enriched inulin (8 g/day for 16 weeks) underwent significant reductions in abdominal body fat deposition and serum levels of triglycerides and IL-6 (involved in MAFLD outcomes); these findings were associated with increases in species of the genus *Bifidobacterium* and decreases in *Bacteroides vulgatus* [51]. Indeed, novel therapeutic approaches for the prevention and management of MAFLD targeting the microbiota include the administration of probiotics, prebiotics, and synbiotics [52].

Of note, dietary components provide nutrients for bacteria utilization, which then produce metabolites putatively implicated in the pathophysiology of MAFLD [53]. The mechanisms underlying hepatic responses to the bioactive substances from gut bacteria (short-chain fatty acids, indole and its derivatives, trimethylamine, secondary bile acids, carotenoids, and phenolic compounds) have been related to the regulation of glycolipid metabolism, immune signaling response, and redox homeostasis [54]. In this regard, total fecal concentrations of microbiome-derived short-chain fatty acids (acetate and propionate) positively correlated with BMI, fasting insulinemia, and insulin resistance in obese women, whose concentrations were significantly reduced after 16 g/d of ITF administration for 3 months in these patients [55].

In addition, polyphenols exert important health benefits due to their antioxidant, anti-inflammatory, and anti-adipogenic properties, being able to modulate the composition of intestinal microbes, which in turn, catabolize polyphenols to release bioactive metabolites [56]. Remarkably, the consumption of grape pomace extract for 4 weeks reduced the serum levels of trimethylamine N-oxide (a metabolite associated with MAFLD severity) in healthy volunteers for reshaping the gut microbiota [57]. Likewise, MAFLD patients treated with litchi-derived polyphenol (oligonol) improved liver steatosis through reducing pathogenic bacteria (*Dorea*, *Romboutsia*, *Erysipelotrichaceae UCG-003*, and *Agathobacter*) and increasing beneficial bacteria (such as *Akkermansia*, *Lachnospira*, *Dialister*, and *Faecalibacterium*) [58]. Moreover, it has been reported that the effects of polyphenols on the regulation of intestinal barrier integrity via induction of transcriptional factors, kinases, and enzymes may account for their positive impact on liver health [59]. Interestingly, changes in microbiome composition (β diversity and specific bacteria) were associated with intrahepatic fat losses after consuming a green Mediterranean diet enriched with green plants and polyphenols for 18 months [60].

In addition, unbalanced intakes of omega 3 fatty acids may affect the gut microbiota homeostasis and alter their absorption, bioavailability, and biotransformation, leading to altered nutrient release to the liver [61]. Therefore, balanced intakes of omega-3 fatty acids may exert a therapeutic role in MAFLD by reverting the altered microbiota profile and increasing the production of anti-inflammatory compounds [62]. For this purpose, the intake of fish oil (rich in omega 3 fatty acids) has been recommended for promoting intestinal wall integrity by interacting with host immune cells [63].

### 1.4. Nutritranscriptomics

The effect of diet on the risk of MAFLD may involve changes in gene expression patterns of critical metabolic genes [64]. For instance, a nutrigenomic study showed that the higher consumption of fructose in patients with MAFLD was accompanied by upregulation of hepatic *TLR4* and *PAI-1* mRNA expression compared to controls, increasing intestinal translocation of bacterial endotoxin [65]. The excessive intake of fructose in patients with MAFLD was also related to a parallel increase in hepatic mRNA expression of fructokinase (an important enzyme for fructose metabolism) and fatty acid synthase (implicated in lipogenesis) genes [66]. In liver cells, *PNPLA3* expression was reversibly suppressed by glucose depletion and increased by glucose refeeding [67]. Of note, dietary polyphenols (resveratrol, quercetin, catechin, and berberine) protected against steatosis in an in vitro model of MAFLD by modulating the expression of mRNA for enzymes participating in lipid metabolism and mitochondrial function [68]. In a similar assay, catechin, quercetin, and resveratrol increased the expression of manganese superoxide dismutase (antioxidant enzyme) and prevented a large increase in the inflammatory TNF-α expression [69].

### 1.5. Nutrimetabolomics

Nutritional metabolomics (nutrimetabolomics) is a key analytical tool that allows the comprehensive metabolic analysis of physiological measurements and energy balance, expediting our ability to identify metabolic diseases that are influenced by foods and nutrients/bioactive compounds to develop targeted diet-based treatments [70]. Thus, nutrimetabolomics focuses on the analysis of many hundreds of metabolites in complex specimens (biofluids, tissues, or cells) to provide better and more individualized biomarkers related to the dietary effects on health and disease [71].

In this context, a systematic review covering the analysis of circulating biomarkers following MAFLD syndrome revealed significant differences in the metabolism of amino acids, fatty acids, and vitamins in MAFLD patients compared to healthy controls [72]. In addition, urinary metabolome testing contributed to defining a metabolic fingerprint associated with MAFLD in obese children by identifying metabolic pathways and metabolites reflecting typical obesity dietary habits and gut–liver axis perturbations [73]. In line with this finding, a Western non-vegetarian dietary pattern positively correlated with a serum metabolite profile characterized by branched-chain amino acids (BCAA), aromatic amino acids, and short-chain acylcarnitines, which in turn, were adversely associated with metabolic alterations including a liver fat measurement [74]. In contrast, plasma alkylresorcinol metabolite, a biomarker for whole-grain intake, was inversely associated with the risk of MAFLD in Chinese adults [75].

Of note, a significant increase in BCAA isoleucine was also identified as a metabolomic salivary signature of obesity-related liver disease in children [76]. Notably, several nutrient-related metabolites (glycocholic acid, taurocholic acid, phenylalanine, and BCAA) increased according to the severity of MAFLD, while glutathione concentrations decreased [77]. Metabolite profiling or serum samples showed that aminotransferase concentrations are a signature of liver metabolic perturbations in MAFLD patients, particularly at the amino acid metabolism and Krebs cycle level [78]. Moreover, serum metabolomic studies found differences in ether- and ester-containing phospholipids as well as in the amino acids lysine, glycine, and isoleucine between obese healthy and obese MAFLD groups [79]. Furthermore, metabolites significantly associated with steatohepatitis in children included methionine, phosphatidylinositol, phosphatidylcholine, sphingolipids, and purine, with relevance in the nutrition field [80].

Regarding micronutrients, the metabolome characterization of human livers showed that vitamin A homeostasis is disrupted in MAFLD and potentially contributes to disease progression, as demonstrated by the hepatic decreased concentrations of retinyl palmitate (RP), all-trans-retinoic acid (*at*RA), 13-*cis*RA, and 4-oxo-*at*RA in these samples [81]. Indeed, altered vitamin A metabolism enhances profibrogenic gene expression, hepatic stellate cell activation, and collagen deposition in the liver, providing insights into cell-specific contributions to vitamin A loss and leading to novel nutritional interventions in liver fibrosis [82]. In a two-stage metabolic tissue screening, hydroquinone and nicotinic acid (NA) were inversely correlated with histological MAFLD severity, whereas only the human nutritional intake of NA equivalent was consistent with a protective effect of this compound against MAFLD progression [83].

## 2. Conclusions

MAFLD is a complex disease, where several endogenous and exogenous factors are involved. Advances in omics technologies are allowing approaches toward multifactorial diseases from a more comprehensive and integral perspective. In this regard, nutrigenetic studies have identified polymorphisms in genes related to lipid metabolism and oxidative stress as associated with the risk of developing MAFLD depending on the dietary consumption of carbohydrates and fats. Selected changes in the expression of lipogenesis-related genes may involve epigenetic mechanisms such as differentially methylated promoter regions, histone acetylation, and specific miRNA induction. Moreover, it has been demonstrated that some nutritional strategies (i.e., prebiotics) reduce liver fat deposition by modulating the gut microbiota and restoring the integrity of the intestinal barrier. In addition, nutrigenomic research has evidenced the upregulation of steatogenic genes after fructose or glucose overloads, which can be reversed by adequate intakes of plant-derived polyphenols (resveratrol, quercetin, and catechin). Furthermore, the biofluid concentrations of several nutrient-related metabolites (such as brain chain amino acids and phospholipids) have been used as biomarkers for the severity of MAFLD. Although more investigation in humans is still necessary, these scientific insights may contribute to gaining a deeper understanding of the molecular and physiological processes underlying MAFLD pathogenesis and phenotype heterogeneity, as well as enabling the characterization of biomarkers of disease progression and severity. This knowledge may also facilitate the identification of therapeutic targets for the implementation of tailored dietary strategies for the prevention, prognosis, and monitoring of MAFLD outcomes through the lens of precision nutrition.

## Figures and Tables

**Figure 1 genes-13-02142-f001:**
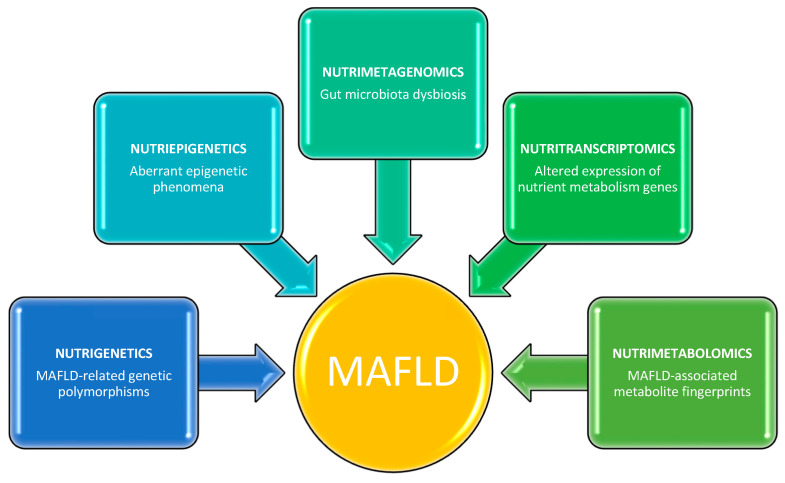
Multi-omics nutritional approaches for identifying potential targets involved in the pathogenesis of MAFLD.

## Data Availability

Not applicable.

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
