# Peer review of "Multi-Omics Nutritional Approaches Targeting Metabolic-Associated Fatty Liver Disease"

_genes, 2022, doi:10.3390/genes13112142_

Round 1

Reviewer 1 Report

The author performed a comprehensive review highlights the potential of nutrigenetic, nutrigenomic, nutriepigenetic, nutrimetagenomic, and nutrimetabolomic approaches for the prevention and management of MAFLD in humans under a precision nutrition scope

-          This reviewer was wondering whether there is a publication bias since the current review is not systematic. Author should pointed this statement as limitation

-          Line 160 - mir-23b, r should be capital;

-          Line 144- to my mind PPARγ not belongs to lipogenic genes;

-          Line 184 – maybe to “higher” abundances – will be correct;

-           Line 214 – via – better in italics;

-          Line 240 – were instead was

Author Response

REVIEWER 1

The author performed a comprehensive review highlights the potential of nutrigenetic, nutrigenomic, nutriepigenetic, nutrimetagenomic, and nutrimetabolomic approaches for the prevention and management of MAFLD in humans under a precision nutrition scope.

-          This reviewer was wondering whether there is a publication bias since the current review is not systematic. Author should pointed this statement as limitation

Response: We agree with the reviewer. To address this issue, the term “literature review” was now added to the manuscript where appropriate.

-          Line 160 - mir-23b, r should be capital;

Response: Thank you. This correction was done.

-          Line 144- to my mind PPARγ not belongs to lipogenic genes;

Response: Thank you for your comment. The term “lipogenesis-related genes” was used by the authors to include SREBP-1c, ACLY, FASN, and PPARγ genes. This term was now included in this sentence: “In vitro assays shown that the tannic acid (a hydrolysable tannin polyphenol found in many dietary plant products such as coffee, tea, cocoa, and sorghum grain) ameliorated lipid accumulation via downregulation of lipogenesis-related genes (SREBP-1c, ACLY, FASN, PPARγ) expression and inhibition of histone acetyltransferase activity in HepG2 human cells [32]”.

Reference

Chung, M.Y.; Song, J.H.; Lee, J.; Shin, E.J.; Park, J.H.; Lee, S.H.; Hwang, J.T.; Choi, H.K. Tannic acid, a novel histone acetyltransferase inhibitor, prevents non-alcoholic fatty liver disease both in vivo and in vitro model. Mol. Metab. 2019, 19, 34-48. doi: 10.1016/j.molmet.2018.11.001.

-          Line 184 – maybe to “higher” abundances – will be correct;

Response: Thank you. The term “higher” was added.

-           Line 214 – via – better in italics;

Response: Thank you. This correction was done.

-          Line 240 – were instead was

Response: Thank you. This correction was done.

Reviewer 2 Report

The review work is well done and very innovative. I recommend improving the figure 1

Author Response

REVIEWER 2

The review work is well done and very innovative. I recommend improving the figure 1.

Response: Thank you for your comment. Figure 1 was improved according to your suggestion and that of another reviewer.

Reviewer 3 Report

Ramos-Lopez chose to review recent scientific evidence on NAFLD/MAFLD associations with nutrigenetics, nutrigenomics, nutriepigenetics, gut microbiome, proteomics, and metabolomics. This is a well-written review on a topic with public health importance as MAFLD prevalence is increasing, but also a much-studied area for multi-omics. Focusing on the nutritional approaches is interesting. I have a few suggestions for the author.

Nutrigenetics paragraph: Considering that most of the genetic section is around PNPLA3, one or two sentences on how much-explained variance attributed to PNPLA3 variants seems in order.

A table summarizing the nutrients associated with multi-omics in MAFLD will be more reader-friendly and highlight the main message.

Nutrigenomics and Nutriepigenetics: I suggest removing all transcriptomics from the Nutrigenomics and Nutriepigenetics, keeping it only Nutriepigenetics, and copying all Nutritranscriptomics to a new section.

Nutrimetagenomics: the author can look at PMID: 33461965 for associations between gut microbiota and NAFLD.

Figure 1: I hope it is a reduced size, otherwise – needs to submit a higher quality figure

Figure 1: Consider adding examples for each of the multi-omics. For example, under “metagenomics”, the association between gut microbiota composition or richness and MALFD (maybe dysbiosis and an arrow for the direction in MAFLD) can be added.

Author Response

REVIEWER 3

Comments and Suggestions for Authors

Ramos-Lopez chose to review recent scientific evidence on NAFLD/MAFLD associations with nutrigenetics, nutrigenomics, nutriepigenetics, gut microbiome, proteomics, and metabolomics. This is a well-written review on a topic with public health importance as MAFLD prevalence is increasing, but also a much-studied area for multi-omics. Focusing on the nutritional approaches is interesting. I have a few suggestions for the author.

Nutrigenetics paragraph: Considering that most of the genetic section is around PNPLA3, one or two sentences on how much-explained variance attributed to PNPLA3 variants seems in order.

Response: Thank you for your comment. This information was now provided in the following sentence: “Interestingly, a meta-analysis demonstrated that people with the GG genotype of the PNPLA3 rs738409 variant were 105% more likely to develop MAFLD, followed by the CG heterozygotes (19% higher risk of MAFLD); conversely, those carrying the CC genotype had 52% lower chance of presenting this outcome [17]”.

The following reference was added:

Salari, N.; Darvishi, N.; Mansouri, K.; Ghasemi, H.; Hosseinian-Far, M.; Darvishi, F.; Mohammadi, M. Association between PNPLA3 rs738409 polymorphism and nonalcoholic fatty liver disease: a systematic review and meta-analysis. BMC Endocr. Disord. 2021, 21, 125. doi: 10.1186/s12902-021-00789-4.

A table summarizing the nutrients associated with multi-omics in MAFLD will be more reader-friendly and highlight the main message.

Response: I understand the point of the reviewer. However, because the information available is still limited, the table would be repetitive of the information provided in the text. I believe that the new Figure 1 better summarizes the content and highlights the importance of multi-omics nutritional approaches in MAFLD.

Nutrigenomics and Nutriepigenetics: I suggest removing all transcriptomics from the Nutrigenomics and Nutriepigenetics, keeping it only Nutriepigenetics, and copying all Nutritranscriptomics to a new section.

Response: Thank you for your suggestion. Nutritranscriptomics was now separated from Nutriepigenetics as a new section.

Nutrimetagenomics: the author can look at PMID: 33461965 for associations between gut microbiota and NAFLD.

Response: Thank you for your comment. The following sentence was now incorporated in the Nutrimetagenomics section: “Interestingly, changes in microbiome composition (beta diversity and specific bacteria) were associated with intrahepatic fat losses after consuming a green-Mediterranean diet enriched with green plants and polyphenols for 18 months [60]”.

The following reference was added:

Yaskolka Meir, A.; Rinott, E.; Tsaban, G.; Zelicha, H.; Kaplan, A.; Rosen, P.; Shelef, I.; Youngster, I.; Shalev, A.; Blüher, M.; et al. Effect of green-Mediterranean diet on intrahepatic fat: the DIRECT PLUS randomised controlled trial. Gut 2021, 70, 2085-2095. doi: 10.1136/gutjnl-2020-323106.

Figure 1: I hope it is a reduced size, otherwise – needs to submit a higher quality figure

Response: Thank you for your comment. I high quality figure was now provided.

Figure 1: Consider adding examples for each of the multi-omics. For example, under “metagenomics”, the association between gut microbiota composition or richness and MALFD (maybe dysbiosis and an arrow for the direction in MAFLD) can be added.

Response: Thank you for your comment. Examples for each of the multi-omics were now provided in Figure 1.